# Evaluation of the Effectiveness of the Use of the Diode Laser in the Reduction of the Volume of the Edematous Gingival Tissue after Causal Therapy

**DOI:** 10.3390/ijerph17176192

**Published:** 2020-08-26

**Authors:** Elisabetta Polizzi, Giulia Tetè, Claudia Targa, Barbara Salviato, Francesco Ferrini, Giorgio Gastaldi

**Affiliations:** 1Department of Dentistry, IRCCS San Raffaele Hospital, Vita Salute University, via Olgettina N.48, 20123 Milan, Italy; polizzi.elisabetta@hsr.it (E.P.); ferrini.f@gmail.com (F.F.); gastaldi.giorgio@hsr.it (G.G.); 2Private Practice, Via Traversagno, 5, 45011 Adria, Italy; clatarga@gmail.com; 3Private Practice, Via Ponte Tresa, 31, 21031 Cadegliano-Viconago, Italy; sal.barbara@icloud.com; 4DDS, Dental School, Vita-Salute San Raffaele University, 20123 Milan, Italy

**Keywords:** plaque-induced gingivitis, gingivitis managing, periodontal disease, diode laser

## Abstract

Aim: The aim of this study was to evaluate and compare two different techniques for the treatment of plaque-induced gingivitis, demonstrating whether the causal therapy supported by diode laser can allow a resolution of the edema caused by gingivitis in less time compared to the single traditional causal therapy. Materials and methods: Twenty-five patients between 20 and 60 years of age with a specific diagnosis of gingivitis were evaluated at the CLID-HSR oral hygiene department. Once the clinical parameters (bleeding index, plaque index, recession, and clinical attack level) were recorded, each of them was subjected to a professional oral hygiene session and instructed in correct home hygiene procedures. Through a split-mouth protocol for each individual patient, hemi-arches were treated by simple randomization to be treated with causal therapy supported by the action of the diode laser (experimental therapy) and which with traditional causal therapy (control therapy). A first intraoral scan was performed before therapy (T0), which was repeated 20 min after rinsing with CHX. The intraoral scans were repeated at a control 7 (T1) and 14 days (T2) after the session. For each intraoral scan, a volumetric value was calculated, proportional to the edema of the gingival tissues, using special digital software. The operator who carried out the volumetric measurements on the software was not aware of the therapy implemented on each half-arch. The operator who carried out the statistical analysis was not aware of the therapy applied to each group. The collected data were statistically compared in order to detect any differences between the volumetric variations between the two therapy groups and within the therapy groups over time. After evaluating the distribution of data by means of the Kolmogorov-Smirnov statistical test, the appropriate nonparametric tests were chosen to carry out the statistical comparisons. Results: Based on the analysis of the gingival-periodontal health parameters and the volumetric value of the treated areas, no statistically significant differences were detected between the areas treated with the adjuvant action of the diode laser compared to those treated with causal therapy alone. Conclusions: With the limitations of this study, in accordance with the statistical results obtained, diode laser therapy does not allow a faster resolution of gingival edema compared to traditional therapy; the two treatment techniques for plaque-induced gingivitis, therefore, have the same efficacy.

## 1. Introduction

Gingivitis is an inflammatory process caused by the presence of bacteria, which, starting from the gingival margin, can also spread to the rest of the gum [1,2,3]. Even when bacterial plaque levels are minimal, the immune system reacts by triggering an inflammatory response that determines the presence of infiltrate [4]. As reported by several studies, sites affected by previous periodontal disease, unlike healthy sites, have a greater predisposition to the onset of gingivitis [5,6,7]. The immune response, in the presence of systemic pathologies of a blood, endocrine, autoimmune and neoplastic nature, may undergo alterations that affect the development of both gingivitis and other oral manifestations of systemic diseases [8,9,10]. The main clinical manifestations of gingivitis are gingival redness, bleeding, and edema [11] not associated with periodontal attachment loss [12]. Since gingivitis does not always cause pain, and the manifestations can be exclusively subclinical, except in rare cases of spontaneous bleeding, an early diagnosis can be difficult [13]. The main cause of the onset of this pathology is considered insufficient oral hygiene [14]; however other possible factors such as smoking [15], genetic alterations [16], systemic diseases of bacterial, viral or fungal origin [17,18,19], autoimmune diseases [20] or of endocrine origin [21], neoplasms [22] and lesions traumatic [23], can promote inflammation of the periodontal support tissues. Although the forms of gingivitis not induced by plaque are not expressly caused by bacterial bioflim, their severity and progression can be influenced by the accumulation of the plaque and the consequent subgingival inflammation [14,24]. The plaque-induced gingivitis treatment protocols provide for the reduction of the etiological agent, that is, the plaque itself, with a consequent progressive improvement of the patient’s oral health conditions, progressively reducing the symptoms associated with the pathology [25,26,27,28]. The aim of this therapy, regardless of the method chosen, is to avoid the progression of bacteria in the subgingival region, involving the connective tissue and the alveolar bone and thus determining the evolution of gingivitis in periodontitis [29,30]. Although the onset of periodontal disease consists of considerable inter-individual variability and depends on various predisposing factors, plaque accumulation represents the main risk factor in the transition between gingivitis and periodontitis [31,32]. The traditional method of treating gingivitis involves the association of mechanical instrumentation with the use of chemical agents with antiseptic action, with the aim, not only of removing the bacterial plaque, but also of guiding the patient in a correct hygienic maintenance therapy, both at home and professionally [33]. Laser treatment has been proposed as a possible therapeutic alternative or in combination with conventional therapy [34,35]. As reported by several studies, the diode laser has provided excellent results in nonsurgical periodontal therapy, demonstrating a bactericidal effect [36,37]. Diode lasers having a wavelength of 980 nanometres and a power of 2 watts have a photodestructive effect, causing the elimination of the periodontal pathogenic microflora [38] Absorption of laser light leads to elevated tissue temperatures, as a result of which most bacteria, including anaerobes, are readily inactivated [39]. Furthermore, compared to traditional mechanical instruments, it allows easier access to areas otherwise difficult to access [40]. The aim of this study was to evaluate and compare two different techniques for the treatment of plaque-induced gingivitis, demonstrating whether the diode laser-assisted causal therapy can allow a resolution of the edema caused by gingivitis in less time than traditional single causal therapy.

## 2. Materials and Methods

### 2.1. Sample Composition

The present pilot study was conducted on a sample of 25 patients, aged between twenty and sixty years. All patients were recruited from the CLID oral hygiene department within the Scientific Hospitalization and Care Institute, San Raffaele Hospital. The sample was made up of specially selected individuals with a specific diagnosis of gingivitis.

The study protocol was approved by the Local Ethics Committee, Milan, no. PR246 and written informed consent was obtained from all study participants in accordance with the Helsinki Declaration.

### 2.2. Inclusion Criteria

In order to participate in the study, patients in good health, free of oral and periodontal systemic diseases, were selected. Left-handed patients were excluded. The whole sample must have completed any drug therapy with antibiotics and anti-inflammatories at least two weeks before participating in the study.

Patients with a specific diagnosis of nonplaque-induced gingivitis were also excluded (1,4,5):Inflammation caused by systemic factors; a typical example is represented by pregnancy or by hormonal upheavals such as puberty or menstruation.Phlogistic-hyperplastic alterations in conjunction with the intake of antiepileptics, calcium channel blockers, and cyclosporine.Gingivitis caused by the use of oral contraceptives.Gingivitis modified by systemic diseases.Gingivitis affecting patients with diabetes mellitus.Gingivitis associated with disorders of the blood picture (e.g., leukemia).Plaque gingivitis modified by malnutrition, such as chronic deficiency of vitamins A, C, and D (although they seem to occur rarely in Europe).

### 2.3. Procedure

The participants were informed about the methods of the experimental study and about the objectives and purposes of the latter. Subsequently, before proceeding, patients had to sign the Informed Consent, in which the protocol was accurately described. Clinical parameters (bleeding index, plaque index, recession, and clinical attachment level) were initially evaluated and recorded in all patients. The standard parameters of plaque index, bleeding index, etc. were evaluated to exclude that the patient could have periodontal disease through the PSR Periodontal Screening and Recording), which consists of a simplified registration of clinical findings. With the PSR, in fact, all the sites of all the dental elements are explored with a periodontal probe, after completing the anamnestic surveys. A first intraoral scan prior to therapy was then performed, which consists of a reference scan; this was repeated at twenty minutes, after having made the patient rinse with 0.2% CHX (T0). Each of them was subjected to the professional oral hygiene session, and through a split-mouth protocol, for each individual patient, they were chosen by simple randomization which hemi-arches to treat with causal therapy assisted by the action of the diode laser (experimental therapy) and which with traditional causal therapy (control therapy). At the end of the session, each individual patient was instructed on correct home oral hygiene procedures. Subsequently, a control scan was performed 7 days after the session (T1) and 14 days (T2), during which the clinical indexes were resumed. For the clinical application of tissue biostimulation, the Raphael Laser (Figure 1), a medical device classified II b (according to Annex IX of the Directive 93/42/EEC).

The diode laser protocol has been the subject of study at our university center, Vita Salute San Raffaele; the study protocol concerns the Raphael diode laser (BIO 980 DMT S.r.l.-Italy: verified and calibrated by the manufacturer has been used in all cases, operating at a wavelength of 940 nm). The laser works continuously with a 3003−3020 μm optical fiber connected to a probe with an output surface of 0.5 cm^2^, capable of emitting a collimated Gaussian beam with very low divergence. The instrument has been positioned at 12−mm perpendicular to the site, thus reducing reflection bias. The movement is defined as “lawn mowing” in the apico-coronal direction. The application time of 80 s for each site and output power set to 2−3.5 W, average fluency of 10 J/cm^2^ and average power density (at the target) was 0.125 W/cm^2^ with a cumulative density dose of 20 J per site for each treatment (max *n*. of treatment 4 × 20 J = 80 J).

For each intraoral scan, the CS 3600 intraoral scanner from Carestream Dental was used by the dentist.

CS 3600 has been designed to acquire 3D still images in the following ways:mandiblemaxillabuccal occlusal registration

The CS 3600 system adopts the following software: Imaging software (CS orthodontic imaging) and CS 3600 acquisition interface. The CS 3600 acquisition interface has made it possible to acquire images in two ways:Partial scan of the arch: several teeth in the preparation area on the mandible and maxilla, and a buccal occlusal registration.Complete scan of the arch: maxilla, mandible and buccal occlusal registration.

### 2.4. Statistic Analysis

The collected data were processed in double-blind since the operator who carried out the static and volumetric analyzes was not aware of the group belonging to the data deriving from the patients. These data were then statistically compared in order to detect any differences between the volumetric variations and through the analysis of the gingival-periodontal health parameters between the two therapy groups and within the therapy groups over time.

The normality of the distribution was tested by Shapiro–Wilk test, and given the non-normality of the data distribution, nonparametric methods were applied. The statistical analyzes were then performed by comparing the two therapy groups with the Mann–Whitney test and the effectiveness of the therapy between T0 and T2 using Friedman’s test.

### 2.5. Treatment of the Collected Data

Once the scans were collected, they were selected to carry out the analysis, the pre-treatment scan (T1), and the scan performed after two weeks (T2).

First, all the detected scans that were presented in STL format were imported into the data management software.

The STL files were imported into a scanning software: Optical Revenge Dental 5.0.

Once imported, both scans obtained simultaneously from the same mouth in half-arches were divided (Figure 2), creating a section that includes the elements of the rear portions.

In this way, the operator obtained four hemi-arches from each individual scan. Once these sections are obtained, the sectoral alignment by hemi-arch is followed so as to be able to proceed with the volumetric analysis. This operation allowed the alignment of two sections of the same subject which takes place automatically (Figure 3) by the software after the operator, having inserted a second scan, gives the command to align the second scan on that previously imported; subsequently, it was also optimized manually (Figure 4) through the choice of landmarks.

In this second phase, hard tissues were used as reference points.

Once superimposed, the images were associated with the same two different colors (Figure 4) to perform a first visual analysis: the most similar files had alternating colors. The files that instead differed totally, or in part, showed big discrepancies, so one color prevailed over the other.

After the alignment procedure, cleaning operations were carried out (mesh cleaning) to delimit the area to be analyzed.

In order to create a solid for each individual impression and perform an evaluation as detailed as possible, the Exocad Model Creator program was used. It allowed to use a fixed surface (Figure 5) as the base of the section, and the impressions, once the cleaning operation was carried out, were positioned parallel to the surface.

This step was made for the sections of both scans so that the two sections presented the exact same distance at the highest point. In this way, the STL files that were previously open have been closed (Figure 6 and Figure 7).

This procedure was necessary to make the two sections completely superimposable in order to continue with the volumetric analysis.

Finally, the files were imported (Figure 8) into the Geomagic Studio 13 software. It was possible to calculate the volume in cubic millimeters, having established the millimeter as the unit of measurement. Having thus had both numerical values available, it was possible to calculate the difference between the two values.

In conclusion, a colorimetric evaluation was performed through a special analysis tool that made it possible to evaluate the volume using the colorimetric scale. This analysis tool used the micron as the reference unit of measurement and made it possible to evaluate the volume of an area or of the whole section through the colorimetric scale (Figure 9) which involves the use of three colors: light blue/blue when the volume is lower, green when the volume is identical for both sections, yellow/red when the volume is higher.

To perform all these evaluations, however, it was necessary that the files were as similar as possible. For this purpose, sections of the scans obtained have been created.

## 3. Results

The collected data were statistically compared in order to detect any differences between the volumetric variations between the two therapy groups and within the therapy groups over time. Statistical comparisons regarding the analysis of PI and BoP parameters are illustrated in Table 1 and Table 2.

Table 1 and Table 2 PI and BOP parameters reported as median, interquartile, mean, and standard deviation of the percentage values and statistical comparison of the two groups. As regards the volumetric analysis, in the same way, there is no statistically significant difference between the volumetric variations of the laser and the control group compared to T0, T1, and T2 (Figure 10, Figure 11 and Figure 12).

After evaluating the distribution of data through the Kolmogorov–Smirnov statistical test, the appropriate nonparametric tests were chosen to carry out the statistical comparisons. Given certain values for each patient, no significant differences were found in the comparison between the two therapy groups. On the basis of the analysis of the gingival–periodontal health parameters and the volumetric value of the treated areas, no statistically significant differences were detected between the areas treated with the adjuvant action of the diode laser compared to those treated with causal therapy alone.

## 4. Discussion

The periodontium includes the following tissues: gingiva, periodontal ligament, root cement, alveolar bone (subdivided into proper alveolar bone and alveolar process) [41]. The gingiva is the portion of chewing mucosa that covers the alveolar process and surrounds the collar of the teeth [42].

Gingivitis is an inflammatory process caused by the presence of bacteria, which starting from the gingival margin, can also spread to the rest of the gum [1,2,3]. The consideration of gingivitis at the implants is called mucositis [43]. Both gingivitis and mucositis, although they are reversible processes, if left untreated, can evolve into an irreversible process of destruction of the periodontal attachment, respectively called gingivitis and peri-implantitis [44,45].

The task of the dentist and dental hygienist is to monitor the patient so as to avoid progression or any recurrence of these pathologies, restoring a state of health of the tooth support tissues and allowing the success of implant-prosthetic rehabilitations over time [46,47,48,49,50].

The intensity of clinical signs and symptoms of gingival inflammation can vary both between different individuals and between sites in the same mouth [31,32]. Normally, the clinical outcomes of plaque-induced gingivitis include erythema, edema, gingival bleeding, and sensitivity [51]. The plaque-induced gingivitis treatment protocols provide for the reduction of the etiological agent, that is, the plaque itself, with a consequent progressive improvement of the patient’s oral health conditions, progressively reducing the symptoms associated with the pathology [33]. Laser treatment has been proposed as a possible therapeutic alternative to traditional mechanical instrumentation associated with the use of chemical agents with an antiseptic action or in association with conventional therapy [34,35].

The diode laser is a semiconductor. The wavelengths of diode lasers most commonly used in dentistry range from 610 to 980 nanometers. It can work in both continuous and intermittent wave (PW) modes. When interacting with the surface of the fabric, the laser light can be refracted, diffused, absorbed, or transmitted. The laser energy absorbed by the tissues can cause heating, coagulation, or vaporization effects depending on the wavelength, power, and optical properties of the tissue [36]. As part of the treatment of periodontal disease, several studies have shown that the use of the laser, if associated with the traditional scaling and root planing method, has provided better clinical and immunological results than conventional therapy [35,52,53,54,55,56,57]. As reported by several authors, since diode laser light is highly absorbed in the hemoglobin of the blood, the laser could represent an effective choice in the removal of highly vascularized inflamed tissues of the periodontal pocket [58]

In reality, compared to what has been shown by the studies mentioned above, the literature is controversial: as reported by several authors, the diode laser, associated with the traditional method, does not provide statistically significant results in microbiological terms and the reduction of clinical signs of gingival inflammation [48,49,50]. In addition, complications such as damage to the supporting tissues due to overheating of the same can be associated with the use of the laser. As reported by several authors, at a temperature above 10 for more than a minute of treatment, damage to the alveolar bone and periodontal ligament can be found, with consequent ankylosis of the dental element and bone resorption [59].

## 5. Conclusions

With the limitations of this study, such as the possible inter-individual variability in terms of compliance, in accordance with the statistical results obtained, diode laser therapy does not allow a faster resolution of gingival edema compared to traditional therapy; the two treatment techniques for plaque-induced gingivitis, therefore, have the same efficacy. Further studies are needed to confirm the results obtained.

## Figures and Tables

**Figure 1 ijerph-17-06192-f001:**
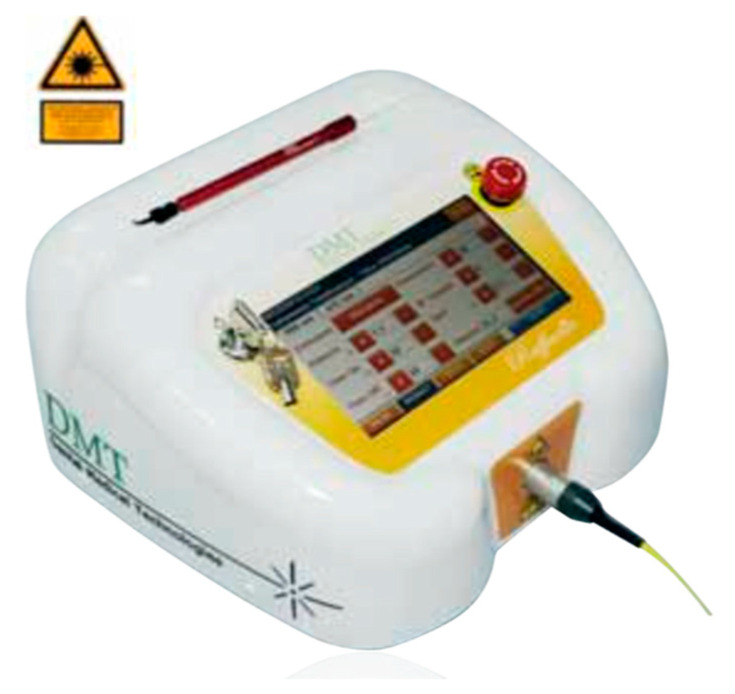
Raffaello diode laser.

**Figure 2 ijerph-17-06192-f002:**
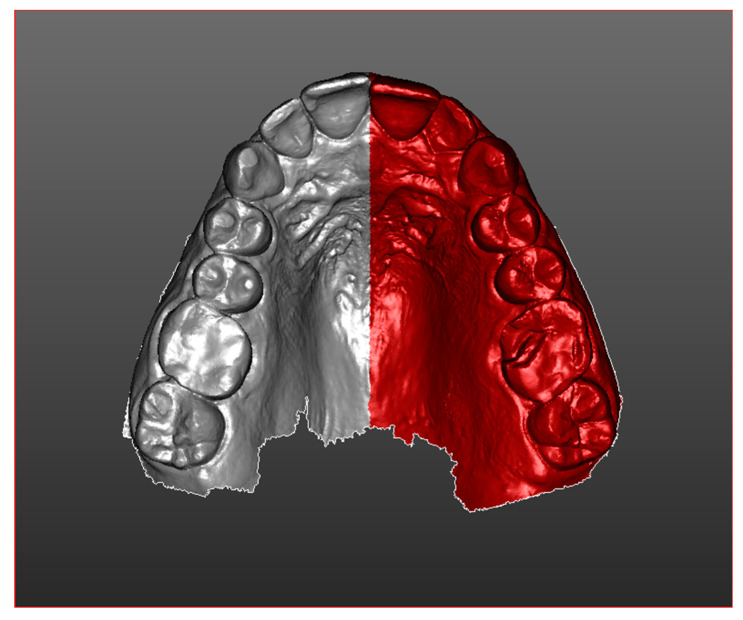
T1 half-arch division.

**Figure 3 ijerph-17-06192-f003:**
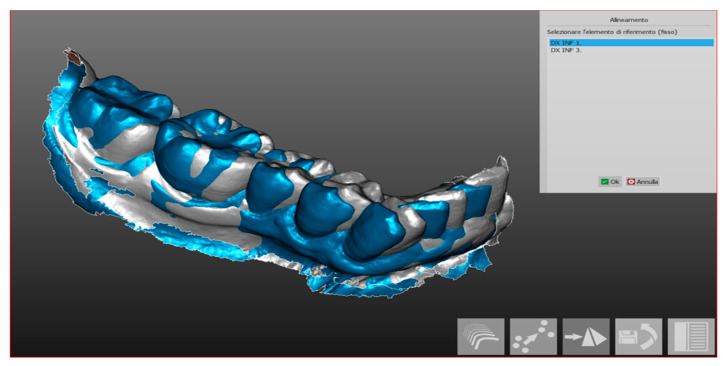
Automatic alignment.

**Figure 4 ijerph-17-06192-f004:**
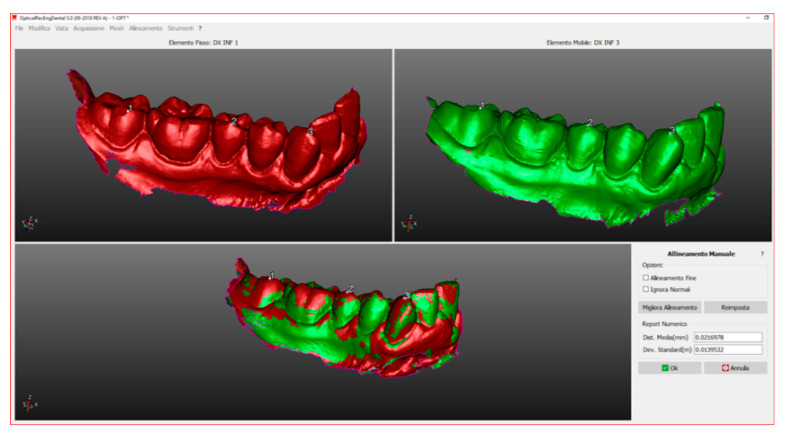
Manual alignment.

**Figure 5 ijerph-17-06192-f005:**
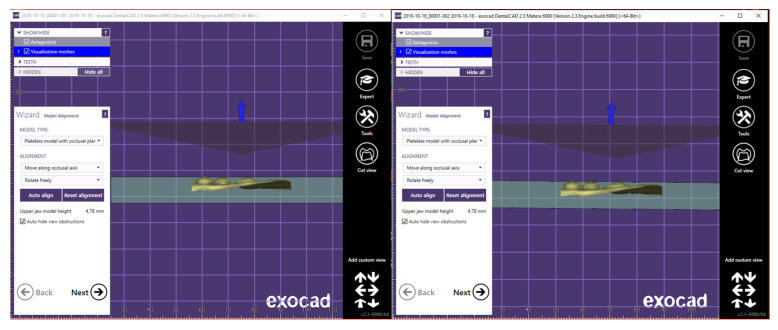
Positioning on the plane.

**Figure 6 ijerph-17-06192-f006:**
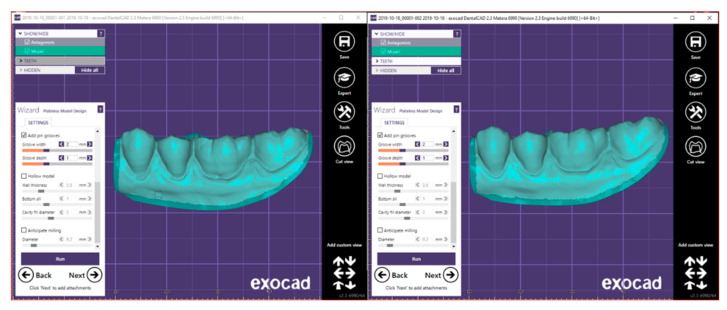
The front section of the solid obtained.

**Figure 7 ijerph-17-06192-f007:**
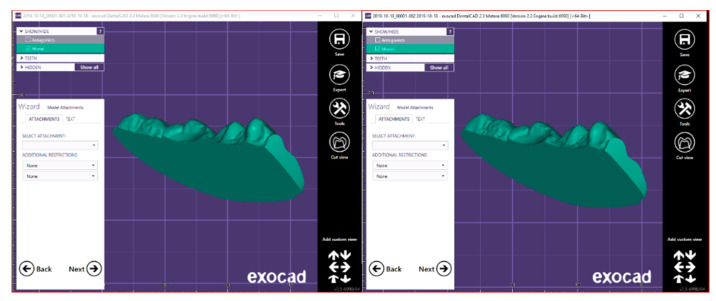
The back section of the solid obtained.

**Figure 8 ijerph-17-06192-f008:**
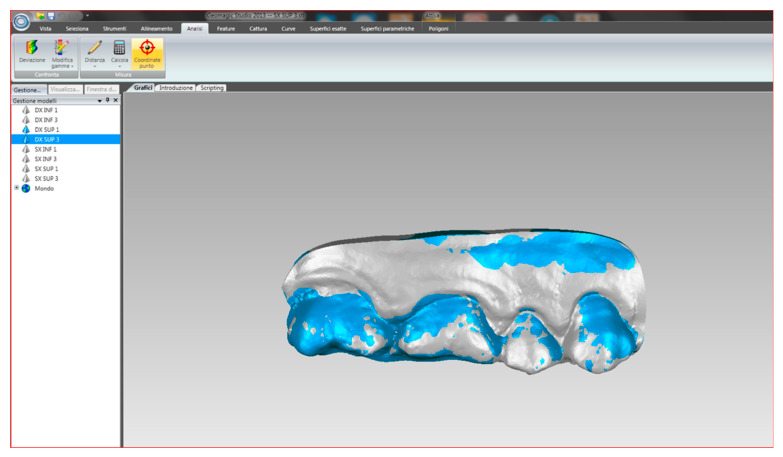
Alignment of the two overlapping solids in the Geomagic software.

**Figure 9 ijerph-17-06192-f009:**
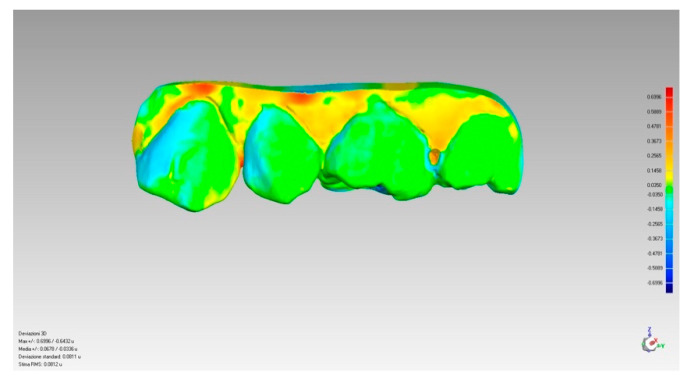
Colorimetric evaluation.

**Figure 10 ijerph-17-06192-f010:**
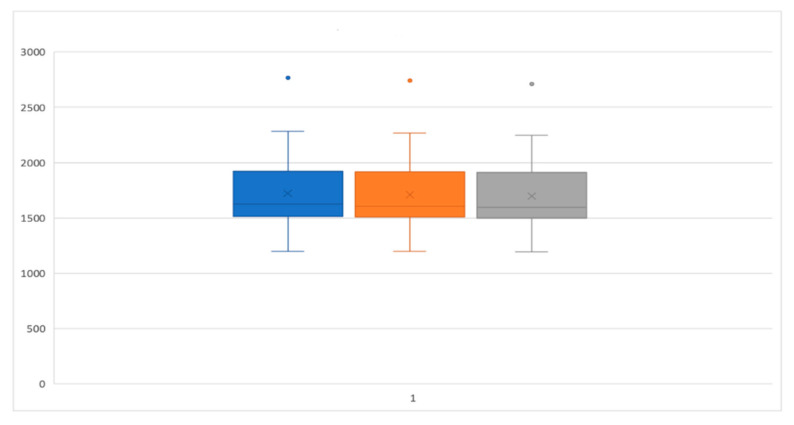
Box and whisker plot representing the performance of the control group at T0, T1, and T2.

**Figure 11 ijerph-17-06192-f011:**
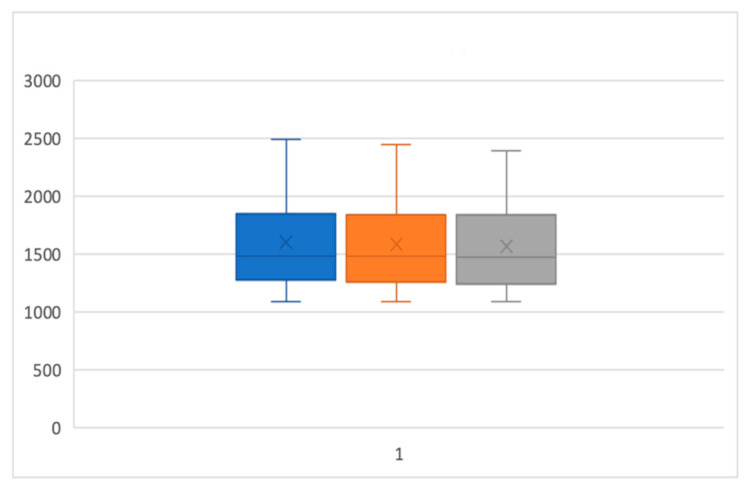
Box and whisker plot representing the progress of the laser unit at T0, T1, and T2.

**Figure 12 ijerph-17-06192-f012:**
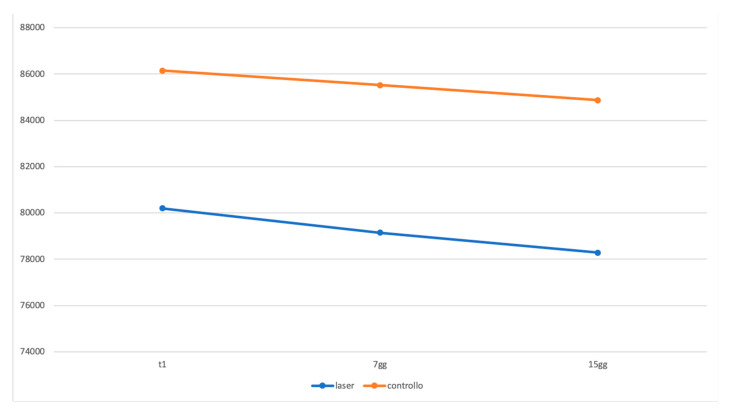
Graphic representation in which the two therapy groups (laser and control) are compared.

**Table 1 ijerph-17-06192-t001:** The collected data of therapy group 1.

Group 1	Procedures	PI MEDIAN (IQR)	BOP MEDIAN (IQR)
T0	LASER	61 (50–82)	33 (28–55)
	CONTROL	60 (48–83)	35 (29–55)
T1	LASER	10 (8–12)	3 (0–6)
	CONTROL	10 (8–12)	3 (1–6)
T1–T0	LASER	−53 (−70–39)	−33 (−50–27)
	CONTROL	−51 (−70–36)	−31 (−49–28)
T2	LASER	10 (9–13)	0 (0–82)
	CONTROL	11 (10–13)	3 (1–6)
T2–T1	LASER	1(0–2)	3 (0–6)
	CONTROL	1(0–1)	0 (0–1)

**Table 2 ijerph-17-06192-t002:** The collected data of therapy group 2.

Group 2	Procedures	PI MEAN (SD)	BOP MEAN (SD)	*p*-Value
T0	LASER	67.32 (22)	46.76 (29.14)	n.s.
	CONTROL	68.48 (22.39)	46.96 (29.09)	
T1	LASER	11.64 (6.68)	4 (5.08)	n.s.
	CONTROL	12.16 (6.54)	4.84 (5.26)	
T1–T0	LASER	−0.56 (0.18)	−0.43 (0.26)	n.s.
	CONTROL	−0.54 (−0.19)	−0.42 (0.25)	
T2	LASER	12.76 (7.57)	67.32 (22)	n.s.
	CONTROL	13.16 (7.40)	5.20 (5.35)	
T2–T1	LASER	0.01 (0.02)	−0.43 (0.26)	n.s.
	CONTROL	0.01 (0.02)	0.00 (0.01)	

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
