# Peer review of "Evaluation of the Effectiveness of the Use of the Diode Laser in the Reduction of the Volume of the Edematous Gingival Tissue after Causal Therapy"

_ijerph, 2020, doi:10.3390/ijerph17176192_

Round 1
Reviewer 1 Report
The aim of the study was to compare the effectiveness of diode laser in the reduction of inflammation as compared with Conventional method in reducing inflammation aS measured by clinical parameters and a scan device. The authors have attempted to reduce bias by having blinded examiners for the different segments of the assessment which is good. however, several clarifications are required in the methodology :
1 as the comparison is intra-individual cross arch - would there be any bias or cross over effect?
2 How effective or credible is the scan device in detecting volume changes? It would be useful to provide some references to support the use of the device
3 rationale of the various assessment time period
The literature review is adequate, some rephrasing and modifications may be required to make the introduction clearer eg line 62 and line 71
Author Response
Review 1
The aim of the study was to compare the effectiveness of diode laser in the reduction of inflammation as compared with Conventional method in reducing inflammation aS measured by clinical parameters and a scan device. The authors have attempted to reduce bias by having blinded examiners for the different segments of the assessment which is good. however, several clarifications are required in the methodology:
Thank you for your suggestion and for appreciating our work. I hope that our answers will be exhaustive.
- As the comparison is intra-individual cross arch - would there be any bias or cross over effect?
- Thank you for your suggestion,
the bias is eliminated using the same protocol for the two therapies and the same subject then the same response from the host alternately and completely randomly for each patient.
- How effective or credible is the scan device in detecting volume changes? It would be useful to provide some references to support the use of the device
- Thank you for your suggestion,
in the text I have underlined in yellow the explanation for which the scanner for the detection of volumetric variations through a specific detection system by sector and by emiarcata and then superimposed calculates with precision the volume variations.
- rationale of the various assessment time period
- Thank you for your suggestion,
there was an error in the description of the timing in the M&Ms that we corrected and highlighted in yellow.
- The literature review is adequate, some rephrasing and modifications may be required to make the introduction clearer eg line 62 and line 71.
- Thank you for your suggestion,
at the base of the two sentences there is a fundamental concept; gingivitis is not only caused by the formation of plaque but also by several factors, because we know that the oral cavity is a continuous interaction between the host and the external environment. periodontal disease is a multifactorial condition, where surely there is also a genetic component, but both more or less serious conditions are compromised by the accumulation of plaque, worsening its course.
Reviewer 2 Report
Please find the comments in following document.

Author Response
- add some references supporting the diode laser use in the treatment of gingivitis
- Thank you for your suggestion,
as you can see in the literature there are not many references in this regard; those that are against the use of the laser for this reason our work reports above all a clinical experience that is the result of many years.
- so T0 was the scan done after the rinse and before the treatment? how can the rinse influence the scan?
- Thank you for your suggestion,
it does not affect the scan simply for safety if two are made, one before rinsing and one after to remove the patient's oral debris.
- it is not clear what exactly are T1 and T3….you previously defined T0 as pre-treatment and T2 as after 14 days, so I suppose that T3 does not exist..be precise in defining the right times
- Thank you for your suggestion,
there was an error in the text; T0 is after the chlorhexidine rinse before therapy; T1 is after therapy; T2 7 days after treatment and T3 14 days of treatment. T2-T3 respect biological tissue healing.
- in this comparison between laser and control group, were IP and BOP considered as a unique variable? it is not clear which variable precisely refers to the p-value
- Thank you for your suggestion,
because both did not give significant results therefore they are considered as variables in consideration at the same time of scanning and in the same patient.
- where are the results about clinical gingival health parameters assessed (the one you previously mentioned in M&M)? it could be interesting also to report the clinical conditions
- Thank you for your suggestion,
the standard parameters of plaque index, bleeding index etc. were evaluated to exclude that the patient could have periodontal disease through the PSR (Periodontal Screening and Recording), which consists of a simplified registration of clinical findings.
With the PSR, in fact, all the sites of all the dental elements are explored with a periodontal probe, after completing the anamnestic surveys.
However we will certainly do other studies to deepen our research.
- report good points and also limitations of the study
- Thank you for your suggestion,
in the conclusions you find what you requested; we can add that the very rigid selection criterion does not allow to have a very large sample, but it also represents a strong point because it minimizes errors.

Reviewer 3 Report
The article is interesting, but the procedure must be clarified according to the objectives The objective has been to compare diode laser treatment and traditional causal therapy on the resolution of edema from gingivitis. Therefore, the protocol must be described in two study groups.
- What has been the protocol to follow in the use of the diode laser? And, in the control group? What is the meaning of traditional causal therapy?
- Which parameters of plaque index, bleeding index, recesion, attachment level have been followed and how were evaluated?
- When interpreting the results of the laser treatment, it is necessary to know the working times, intervals in the application, joules and hertz,...What is the wavelength of the laser that the authors have used?
- For the control group, the concept of traditional causal therapy must be described
Introduction and discussion: the authors must manage the literature in relation to the effective treatment, laser and "traditional"
Author Response
Review 3
The article is interesting, but the procedure must be clarified according to the objectives The objective has been to compare diode laser treatment and traditional causal therapy on the resolution of edema from gingivitis. Therefore, the protocol must be described in two study groups.
Thank you for your suggestion, however, the difference between the two protocols was not stressed because it was using a common protocol that differed only in the final instrument being both carried out on the individual patient, so the basic treatment protocol is the same to eliminate bias. The final objective is the resolution of gingivitis and the criteria of inclusion of patients have been very selective in order to reduce bias.
- What has been the protocol to follow in the use of the diode laser? And, in the control group? What is the meaning of traditional causal therapy?
- The diode laser protocol has been the subject of study at our university center, Vita Salute San Raffaele; the study protocol concerns the Raphael diode laser (BIO 980 DMT S.r.l.-Italy: verified and calibrated by the manufacturer has been used in all cases, operating at a wavelength of 940 nm). the laser works continuously with a 300-320 μm optical fiber connected to a probe with an output surface of 0.5 cm2, capable of emitting a collimated Gaussian beam with very low divergence. The instrument has been positioned at 1-2mm perpendicular to the site, thus reducing reflection bias. the movement is defined as "lawn mowing" in the apico-coronal direction. The application time of 80 s for each site and output power set to 2-3,5 W, average fluency of 10 J/cm2 and average power density (at the target) was 0.125 W/cm2 with a cumulative density dose of 20 J per site for each treatment (max n. of treatment 4×20J = 80 J).
While for the traditional protocol a traditional causal therapy or oral hygiene session was performed. The significance of non-traditional causal therapy is to eliminate the very cause of gingivitis in this case plaque.
- Which parameters of plaque index, bleeding index, recesion, attachment level have been followed and how were evaluated?
- Thank you for your suggestion,
the standard parameters of plaque index, bleeding index etc. were evaluated to exclude that the patient could have periodontal disease through the PSR Periodontal Screening and Recording), which consists of a simplified registration of clinical findings. With the PSR, in fact, all the sites of all the dental elements are explored with a periodontal probe, after completing the anamnestic surveys.
- When interpreting the results of the laser treatment, it is necessary to know the working times, intervals in the application, joules and hertz,...What is the wavelength of the laser that the authors have used?
- Thank you for your suggestion,
however, we have reported the entire protocol in response to your first question with wavelength and application time.
- For the control group, the concept of traditional causal therapy must be described
- Thank you for your suggestion,
however, this step too was well defined and underlined in its key point in response to his first question.
Round 2
Reviewer 2 Report
Line 127: Please correct times definition, if you say that T3 is after 14 days it can not be T2. Please check all the text about times definition.
Line 219: Answer to one comment was not given: "Please define IP" in the text and in the tables.
Author Response
- thanks for the suggestion,
there was an error in the timing T3 is not there, they arrive precisely up to T2 i.e. 14 days after the oral hygiene session. we highlighted in text - thanks for the suggestion,The plaque index (PlI) (Silness J & Löe H) is recorded during the periodontal clinical examination at 6 sites for each tooth element present by circumferential probing with a manual periodontal probe. In this way the clinical periodontist obtains an accurate assessment of the amount of bacterial plaque not removed.
IP was misspelled in the text without translating that English would be PI. we correct and highlighted in text

Reviewer 3 Report
The guidelines for laser application must be included in the text, in the material and method section. In addition, the parameters of the gingival and periodontal disease studied must also be included.
Author Response
Thanks for the suggestion,
we have inserted all the required and underlined parts in yellow in the text.
